# Exploring Group and Symmetry Principles in Large Language Models

## Abstract

Large Language Models (LLMs) have demonstrated impressive performance across a wide range of applications; however, assessing their reasoning capabilities remains a significant challenge. In this paper, we introduce a framework grounded in group and symmetry principles, which have played a crucial role in fields such as physics and mathematics, and offer another way to evaluate their capabilities. While the proposed framework is general, to showcase the benefits of employing these properties, we focus on arithmetic reasoning and investigate the performance of these models on four group properties: closure, identity, inverse, and associativity. Our findings reveal that LLMs studied in this work struggle to preserve group properties across different test regimes. In the closure test, we observe biases towards specific outputs and an abrupt degradation in their performance from 100% to 0% after a specific sequence length. They also perform poorly in the identity test, which represents adding irrelevant information in the context, and show sensitivity when subjected to inverse test, which examines the robustness of the model with respect to negation. In addition, we demonstrate that breaking down problems into smaller steps helps LLMs in the associativity test that we have conducted. To support these tests we have developed a synthetic dataset which will be released.

## 1 Introduction

Large Language Models (LLMs) have shown remarkable capabilities across various domains, serving as the foundation for numerous applications Anil et al. (2023); Bubeck et al. (2023); OpenAI (2023); Brown et al. (2020); Kojima et al. (2022); Chowdhery et al. (2022); Saparov & He (2022).

Additionally, several studies have explored various aspects of the reasoning capabilities of LLMs showing significant performance improvements compared to previous domain specific models Kojima et al. (2022); Wang et al. (2022); Mukherjee et al. (2023); Mitra et al. (2023); Shen et al. (2023). These studies have highlighted LLMs being able to understand and solve complex problems. Despite comprehensive evaluation on various benchmarks, LLMs still might fail in unpredictable ways. In this work, we propose a simple yet helpful framework to study LLMs behavior from a different perspective grounded in group and symmetry principles.

Group and symmetry principles which made significant contributions to numerous fields, such as physics, mathematics, and chemistry Sarlet & Cantrijn (1981); Jaffé & Orchin (2002); Hargittai & Hargittai (2009); Fujita (2012); Gazizov & Ibragimov (1998); Weyl (2015); Polak & Trivers (1994). These principles are foundation for understanding the fundamental structure of laws and the behavior of complex systems. For instance, in physics, group and symmetry principles have played a pivotal role in shaping modern physics by offering profound insights into the fundamental structure of physical laws and the behavior of physical systems. Noether's theorem serves as a prime example, stating that every differentiable symmetry of the action of a physical system with conservative forces has a corresponding conservation law Kosmann-Schwarzbach et al. (2011); Brading & Brown (2003). As another example, temporal translation symmetry states that the laws of physics remain unchanged over time. This invariance of physical laws under time translation leads to the conservation of energy. This principle is essential to our understanding of various physical phenomena and is widely applied across numerous disciplines, including mechanics, thermodynamics, and electromagnetism Gildener (1976); Aharonov et al. (1964); Sasa & Yokokura (2016); Lewis (1930). Other group and symmetry principles in physics, including translation, rotation, and gauge symmetries, have contributed to the

discovery of conservation laws and the formulation of fundamental theories like the Standard Model of particle physics Weinberg (1969); Feynman & Weinberg (1999). These principles have also been instrumental in understanding the behavior of complex systems, such as condensed matter systems and cosmological models Zee (2010); Anderson (1972).

Exploring the role of group and symmetry principles in LLMs can lead to:

- Alternative Perspective to Measure LLMs' Performance: Investigating the role of symmetry in LLMs can provide new strategies for evaluating their performance, potentially leading to better learning methods. The significance of symmetry principles and group properties in these domains can be exemplified by the Winoground task and dataset Thrush et al. (2022), which evaluates the ability of vision and language models to conduct visio-linguistic compositional reasoning by matching images with captions containing identical sets of words in different orders. For example, given an image and two captions, *'an old person helping a young person'* and *'a young person helping an old person'* a model that understands the non-symmetric nature of these sentences and leverages group properties can provide the correct answer Thrush et al. (2022); Lin et al. (2023). By connecting the role of symmetry and group properties in LLMs to the reasoning capabilities required for tasks like Winoground, we can develop a more comprehensive understanding of model performance and devise improved learning methods.

- Insights for Interpretability: Studying group properties and symmetry principles in LLMs can lead to more interpretable models, similar to the insights gained from saliency maps. This method is based on the idea that if a minor perturbation in the input does not alter the meaning but results in a different output, the perturbed portion of the input is crucial for the model's prediction Simonyan et al. (2013); Sundararajan et al. (2017); Zeiler & Fergus (2014). These principles help uncover hidden relationships and structures within the data, thereby facilitating improved decision-making and trust in the model's predictions. For example, in an identity test, introducing various symmetries, such as inverse symmetry, to perturb the input query causes the output result to change. This is similar to obtaining a saliency map by perturbing the prompt query and feeding it to the LLM. While saliency maps and symmetry principles share similarities in their focus on understanding the importance of different input elements, their approaches differ. Saliency maps rely on perturbations to identify crucial input components, whereas symmetry principles and group properties delve into the inherent structure and relationships within the data. Group properties investigate the algebraic structures governing the interactions between data elements, such as the presence of an identity element, the existence of inverses, and the associative property. In addition, symmetry principles examine how certain aspects of the data remain unchanged or exhibit similar patterns under specific transformations, including rotations, reflections, and translations. By combining these approaches, a more comprehensive understanding of the data's underlying organization can be achieved.

- Robustness, Reliability: Analyzing the influence of group properties on LLMs can aid in designing ensemble strategies that combine the predictions of multiple LLM's outputs, to enhance overall robustness and generalization. Additionally, it can help to better examine the confidence of LLM's output. For instance, using the closure, inverse, identity, and associativity tests, we observe that LLMs studied in this work fail for sequences longer than 15 elements, and the confidence level decreases as we increase the length of the sequence in the simple addition task.

- Generalization across Domains: Incorporating symmetry and group properties into LLMs can improve their generalization capabilities across various domains such as chemistry. For example, for a language model that is trained to predict the products of a reaction $A + B \rightarrow C + D$, by understanding that the reaction $B + A$ should have a similar prediction as $A + B$ due to the symmetric nature of the reactants, the model can leverage this knowledge to improve its performance on reaction prediction tasks. The understanding of symmetry and group properties in this case allows the language model to recognize the invariance of reaction outcomes under different reactant orders, leading to more consistent predictions of chemical reactions.

Our primary focus in this study is to assess performance of LLMs in arithmetic reasoning tasks using these principles. Our goal is NOT to show that they have good or bad performance in arithmetic operations as this has already been studied in previous studies Kojima et al. (2022); Mitra et al. (2023); Wang et al. (2022), but to use them as scenarios to show the added value of using group properties for analysis. We investigate the performance of GPT-4 and GPT-3.5 on four group properties: closure, identity, inverse, and associativity. Our main findings are:

- Language models studied in this work exhibit significant sensitivity to the identity test. For instance, GPT-4 achieves a $100\%$ accuracy rate in the closure test for sequences involving the addition of ones, with lengths varying from 5 to 35. However, when we assess the identity test by incorporating zeros into the input, even at a smallest proportion of $25\%$ of the sequence length, GPT-4's accuracy experiences a considerable decline, approaching zero.

  A similar pattern is observed in GPT-3.5. A good analogy for the identity test is introducing irrelevance to the input of natural language. Consider the following example from the GSM-IC dataset Shi et al. (2023):

  > Original Problem:
  > Q: Elsa has 5 apples. Anna has 2 more apples than Elsa. How many apples do they have together?
  > Problem with Irrelevant Context:
  > Q: Elsa has 5 apples. Anna has 2 more apples than Elsa. Liz has 4 peaches. How many apples do they have together?

  The authors in Shi et al. (2023) demonstrate that even with a variety of prompting techniques on the GSM-IC dataset the models are all sensitive to irrelevant information in the input. By leveraging group properties, specifically identity test, we can assess the robustness of the model in a more controlled and cost effective manner.

- In the identity test, we observed that the accuracy of models such as GPT-4 and GPT-3.5 decreases when zeros are inserted into the summation of ones. The performance decline follows this order: adding irrelevant information randomly, insertion in the middle, adding irrelevant information at the beginning, and lastly, at the end. Our results align with Liu et al. (2023), which demonstrated that changing the position of relevant information within the input context of language models leads to variations in performance. This finding suggests that introducing irrelevant elements at various positions within natural language sequences can pose a significant challenge for models like GPT-4 or GPT-3.5. However, employing group properties and simple tests, such as addition, can aid in validating and identifying these issues early on with less computation costs.

- In our analysis of the inverse properties within group properties, we found that the LLMs investigated in this work display sensitivity when subjected to inverse tests. Our inverse test consists of adding negative ones to the sum of ones, yielding a final result of zero. Inverse properties bear resemblance to the introduction of negation in natural language. Similarly, the study by Truong et al. (2023) highlights that LLMs exhibit multiple limitations in handling negation, including an inability to reason effectively under negation. By implementing the test framework proposed in this study, these issues can be identified in advance in a given LLM.

- In the associativity test, we decompose each summation into smaller steps for the model to process. Our findings indicate that GPT-4 demonstrates a significant improvement in performing addition tasks compared to GPT-3.5 when problems are broken down into smaller components. Our results highlight the importance of decomposing problems into smaller, more manageable pieces that can be effectively solved by the model which is reported using more complex tasks in Kojima et al. (2022). By employing group analysis, it is possible to identify and address these issues through the implementation of straightforward tests proposed in this work.

## 2 SYMMETRY IN LLMS

Symmetry and invariance are fundamental concepts in understanding the behavior of systems Goodman et al. (2009). An object or quantity is said to be invariant if it remains unchanged under transformations Kosmann-Schwarzbach et al. (2011). Consider an arbitrary quantity $F = F(A, B, C, ...)$ that depends on different quantities. If we transform $A, B, C, ...$ to their respective primed variables $A', B', C', ...$, and we have

$$F(A', B', C', ...) = F(A, B, C, ...) \tag{1}$$

then $F$ is said to be invariant under the given transformation.

Invariance can also be described through the concept of symmetry. A system is symmetric if it remains the same after a transformation or class of transformations. For instance, a physical system

is symmetric under rotations if it can be rotated in any direction and remains unchanged. The set of all transformations that leave a given object invariant is called a symmetry group Olver (1995); Goodman et al. (2009).

Throughout this paper, we will use the following symmetries to investigate the capabilities of LLMs:

**Translation Symmetry**: This type of symmetry refers to the invariance of a system or pattern under a spatial transformation, such as shifting or sliding. In the context of LLMs, translation symmetry can help us understand how the models respond to changes in the position of elements within a sequence Weyl (2015).

**Random Swapping Symmetry**: Also known as permutation symmetry, this concept involves the invariance of a system or pattern under the exchange of its elements. In our study, we will use random swapping symmetry to analyze how LLMs handle rearrangements of input elements, providing insights into their robustness and generalization capabilities French & Rickles (2003).

**Inverse Symmetry**: This type of symmetry is characterized by the invariance of a system or pattern under an operation that reverses its elements or their order. In the context of LLMs, we will explore how the models perform when presented with inputs that have been transformed using inverse symmetry, shedding light on their ability to recognize and process different representations of the same information Morandi et al. (1990).

To better understand the implications of these symmetries, we can examine them within the context of group theory. A group is a set $G$ and binary operator '∘' that satisfies closure, identity, inverses, and associativity Aschbacher (2000); de La Harpe (2000). A group is a set $G$ and operator ∘ such that:

- Closure: $G$ is closed under ∘; i.e., if $a, b \in G$, then $a \circ b \in G$.
- Identity: There exists an identity element $e \in G$; i.e., for all $a \in G$ we have $a \circ e = e \circ a = a$.
- Inverses: Every element $a \in G$ has an inverse in $G$; i.e., for all $a \in G$, there exists an element $a' \in G$ such that $a \circ a' = a' \circ a = e$.
- Associativity: The operator ∘ acts associatively; i.e., for all $a, b, c \in G$, $a \circ (b \circ c) = (a \circ b) \circ c$.

Group principles are essential in the laws of nature, as they encapsulate regularities that remain consistent regardless of specific dynamics. These principles impart structure and coherence to natural laws, enabling a better understanding of physical events and the discovery of the laws themselves. Schwichtenberg (2018); Lax (2001); Tung (1985); Bishop (1993); Golubitsky & Stewart (2003).

Our goal is to conduct a comprehensive analysis for a given LLM to determine whether they maintain group properties and symmetry principles. In this investigation, we will cover a broad spectrum of symmetries, including but not limited to rotational, translation, scaling, swapping, and inverse symmetries. Examining group properties can help us understand the fundamental characteristics of the LLMs and potentially improve their performance in reasoning tasks due to the following factors:

- Identifying patterns: By leveraging group properties, LLMs can effectively identify and analyze underlying patterns and relationships within a problem, enhancing their problem-solving capabilities. For instance, the set of even numbers can be considered as a group, which enables an LLM to unveil underlying connections and patterns among them. For instance, it can determine that the sum of two even numbers is always even. By recognizing these patterns, the LLM can make accurate predictions and solve problems related to number sets and arithmetic operations more efficiently.

- Comprehending problem structure: Investigating group properties can potentially contribute to address complex reasoning tasks more efficiently. For instance, recognizing the group properties of integers under addition operator allows LLMs to devise strategic approaches to solve arithmetic tasks. When solving for the sum of a series of integers, the LLM can employ associativity to rearrange terms, thereby simplifying the problem. Additionally, recognizing the role of identity elements and inverses can help the LLM quickly identify shortcuts or eliminate unnecessary calculations, leading to more efficient problem-solving and accurate results.

- Reliable solutions: The utilization of group properties in reasoning tasks contributes to the generation of solutions that are both accurate and reliable, ensuring high-quality outcomes. For instance, if an LLM preserves the group properties of addition on a list of integers, it suggests that the LLM has a good understanding of the concept of addition operation.

Our contributions in this work are:

- An alternative evaluation perspective: Symmetry and group principles as tools for evaluating and understanding LLMs offer a different angle for evaluation, and can provide new insights about the inner workings of LLMs, helping to identify their strengths and weaknesses.
- Dataset: We have developed a synthetic dataset specifically designed to evaluate the performance of LLMs in terms of group principles. This dataset allows us to systematically test LLMs' abilities to in terms of group principles.

## 3 EXPERIMENTS

### 3.1 GROUP OF ADDITION ON A LIST OF INTEGERS

Throughout the paper, we use the addition operator as the running example example to demonstrate our framework. We evaluate whether a given LLM maintains the group properties of integer addition. We propose the following experimental approach.

> **Closure Test**
> 1. Create a list of integers, such as [1, 2, 3].
> 2. Calculate the sum of the list, which in this case is 6.
> 3. Apply a symmetry operation to the list, such as swapping the first and last elements. This results in a new list like [3, 2, 1].
> 4. Request the LLM to calculate the sum of the modified list.
> 5. If the LLM consistently yields the same integer result as the original list (6 in this example), it implies that the LLM preserves the closure property of the group.

> **Identity Test**
> 1. Append zeroes to the list, resulting in a new list like [1, 2, 3, 0, 0].
> 2. Calculate the sum of the new list, which should be the same as the original list (in this case, 6).
> 3. Perform a symmetry operation on the new list, such as reversing it to get [0, 0, 3, 2, 1].
> 4. Ask the LLM to compute the sum of the modified list.
> 5. If the LLM consistently produces the same sum as the original list (6 in this example), it suggests that the LLM preserves the identity property of the group.

> **Inverse Test**
> 1. Generate a new list where each integer is replaced by its negation, resulting in a list like [-1, -2, -3].
> 2. Perform a symmetry operation on the new list, such as swapping the first and last elements to get [-3, -2, -1].
> 3. Ask the LLM to compute the sum of the two lists ([1, 2, 3] and [-3, -2, -1]).
> 4. If the LLM consistently produces a sum of 0, it suggests that the LLM preserves the inverse property of the group.

> **Associativity Test**
> 1. Split the list into sublists in different ways, for example, [[1], [2, 3]] and [[1, 3], [2]].
> 2. Perform a symmetry operation on each sublist, such as reversing each sublist. This results in new arrangements like [[1], [3, 2]] and [[3, 1], [2]].
> 3. Ask the LLM to compute the sum for each arrangement.
> 4. If the LLM consistently produces the same result for all arrangements (6 in this example, which is the sum of the original list), it suggests that the LLM preserves the associativity property of the group.

These tests ensure to check if a given LLM not only understands the concept of addition and the associated group properties but also the symmetries inherent in the operation.

### 3.2 RESULTS

We present the results of our experiments on the summation group. The summation of integer values forms a group because the sum of any two integers is also an integer, thereby satisfying the closure property. The identity element in this group is zero, and the inverse of a value is its negative counterpart. Furthermore, the summation of integers preserves the associativity property. To assess the group properties using the summation operator, we work with a set of simple arithmetic expressions containing only the elements one, zero (serving as the identity element), and negative one (acting as the inverse element). We then proceed to investigate all the group properties within these

expressions. Initially, we compared our findings with other open-source language models, such as LLAMA-2. However, the performance of these models was significantly lower than that of GPT-3.5 and GPT-4-32k, leading us to concentrate on the latter for our analysis. For a more detailed exploration of our experiments with small language models (SLMs), please see Appendix 5. Furthermore, we conducted experiments by introducing irrelevant data to models like GPT-4o and Mistral-7B-Instruct across 200 randomly selected test questions from the GSM8K dataset. Additionally, a simple example of group theory and symmetry in vision models is provided in Appendix 5. We have conducted each experiment 10 times and have reported the average accuracy.

### 3.3 CLOSURE TEST

For the closure test, we have examined a set of elements consisting solely of ones, combined using the summation operation. We have created sets of varying lengths and test expressions ranging from a summation of five ones $(1 + 1 + 1 + 1 + 1)$ to a sum of 150 ones.

The results are presented in Figure 1. The x-axis displays varying lengths of expressions consisting of repeated ones in summation, while the y-axis represents the accuracy of the two LLMs, GPT-4-32k and GPT-3.5. The color indicates the average accuracy over 10 runs for each test. It becomes evident that GPT-3.5 and GPT-4-32k provide accurate results for sums of ones up to 35 elements; however, their performance declines beyond this point, failing to maintain closure properties. This observation suggests that LLMs' capabilities for performing summation are significantly limited beyond a certain sequence length. Another insight from Figure 1 is that these LLMs can accurately compute sums for 50 and 100 ones. However, further analysis reveals that they predominantly return values of 100 and 50 when the actual values are *around and not exactly* 100 and 50, respectively, indicating a significant bias towards these values. Figure 2 showcases the frequency of GPT-4-32k outputting 100 (blue) and 50 (red) when the ground truth ranges from 5 to 150 for closure test expressions which represents a bias towards these values. Additionally, this experiment demonstrates that, due to these biases, repeating the experiment will not improve the results. For the ablation study, please refer to Appendix 5.

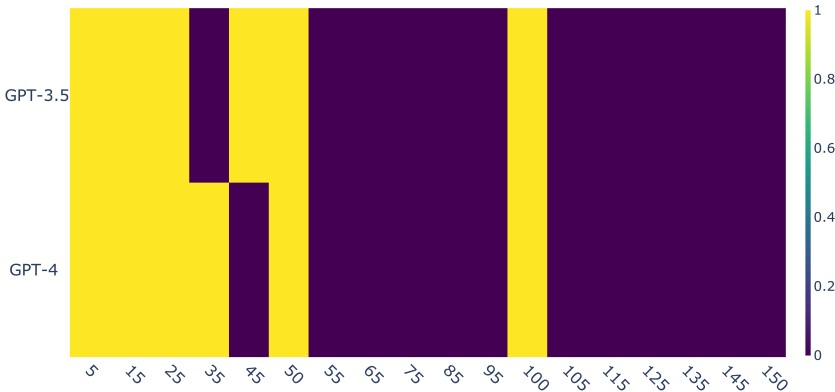

Figure 1: *Closure test:* Average accuracy of GPT-4-32k and GPT-3.5 for sums of ones. The x-axis illustrates the varying lengths of expressions composed of summations of repeated ones. The y-axis denotes the accuracy of the two LLMs, GPT-4-32k and GPT-3.5. The color represents the average accuracy obtained from 10 runs for each test.

### 3.4 IDENTITY TEST

In our identity test, zero is the identity element for the summation of integers of ones. For this test, we will first add different proportions of zeros to our summation expressions. We will choose the ratios of $[0.25, 0.5, 0.75, 1]$ for adding zeros with respect to the expression length. For example, for the ratio of 0.5 and the expression $1 + 1 + 1 + 1 + 1$, we will modify it to $1 + 1 + 1 + 1 + 1 + 0 + 0$.

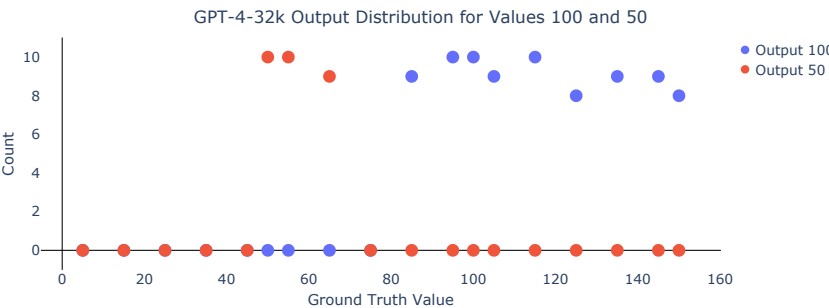

Figure 2: Number of times GPT-4-32k outputs 100 (blue) and 50 (red) compared to ground truth for closure expressions. This visualization emphasizes the biases in the LLMs' responses and offers a deeper insight into their limitations when handling summation tasks.

By passing these expressions, we can investigate if and to what extent LLMs studied in this work can preserve the identity test.

In our experiment, we can also apply different symmetries to each expression. We apply inverse, random swapping, and translation symmetry. For the inverse symmetry, we will place zeros from the end of the sequence to the beginning, and for the translation symmetry, we will shift all the zeros to the middle of the expressions. For random swapping, we will swap the zeros randomly. Then, we provide these expressions as input to the LLMs. For example, for the expression $1 + 1 + 1 + 1 + 1$, we pass the following expressions to LLMs to examine their identity preservation test:

- $1 + 1 + 1 + 1 + 1 + 0 + 0$ (adding identity elements)
- $0 + 0 + 1 + 1 + 1 + 1 + 1$ (inverse symmetry)
- $1 + 0 + 1 + 1 + 1 + 1 + 0$ (random swapping symmetry)
- $1 + 1 + 0 + 0 + 1 + 1 + 1$ (translation symmetry)

Figure 3 presents the results of these expressions with different lengths ranging from 5 to 150 and displays the accuracy of each LLM on each of these tests with different ratios of adding zeros. Altering the expressions using different symmetries changes the results and deteriorates the accuracy in many cases. The accuracy of these models degrades for various symmetric expression variations beyond a sequence length of 5.

3.5 INVERSE TEST

To test the inverse properties of a group, we add the negative values of each expression to the expression itself and examine different variations of the expression. We then test different modifications of the expression using symmetry principles, such as inverse, random swapping, and translation symmetry. For inverse symmetry, we add the inverse element to the beginning of the expression. For translation symmetry, we add the inverse element to the middle of the expression. For random swapping, we add the inverse elements to random positions within the expression. For example, to test the inverse properties, we modify the expression 1+1+1+1+1 to the following variations, with the expectation that the ground truth for each expression should be zero:

- $1 + 1 + 1 + 1 + 1 - 1 - 1 - 1 - 1$ (adding inverse elements)
- $-1 - 1 - 1 - 1 - 1 + 1 + 1 + 1 + 1 + 1$ (inverse symmetry)
- $1 + 1 - 1 + 1 - 1 - 1 + 1 - 1 - 1 + 1$ (random swapping symmetry)
- $1 + 1 + 1 - 1 - 1 - 1 - 1 - 1 + 1 + 1$ (translation symmetry)

There is a caveat here: if LLMs are biased towards zeros, this might affect the results. However, this is not a significant concern, as we are not relying solely on this experiment to understand how inverse properties impact these models. We could test these expressions by changing them in a way that the summation is non-zero, for example to 1+1+1+1+1-1-1 or 1+1+1+1+1-1-1. However, for simplicity, we will only consider adding the same number of negative ones to each expression in this experiment. Figure 4 shows the result of our experiment for the inverse properties. We can observe from the figure that the LLM's outputs for some of these expressions fail early on, while for others, it returns

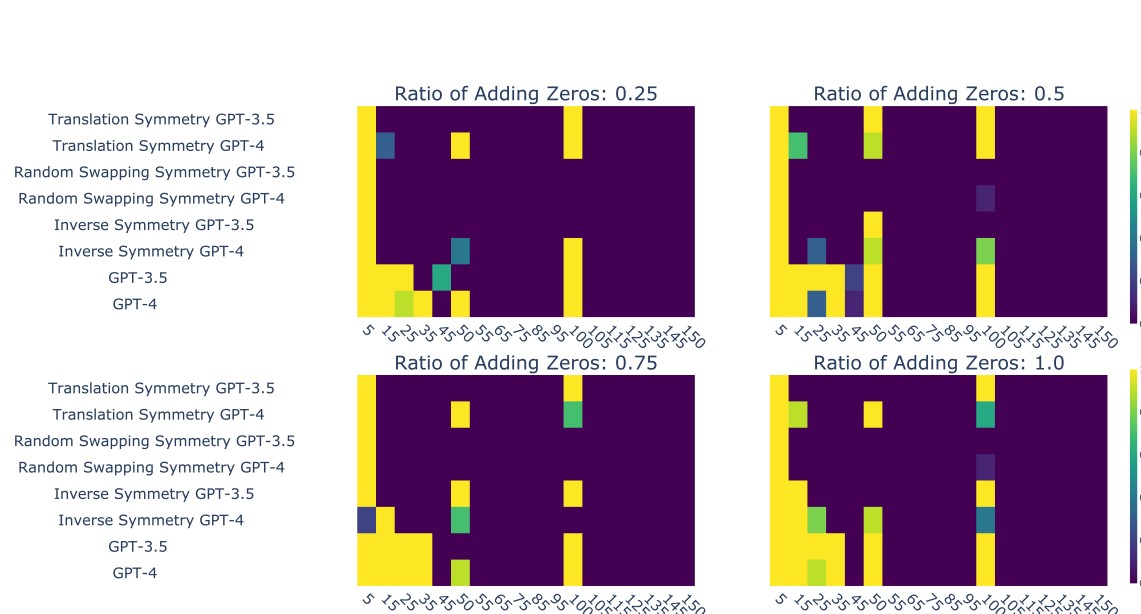

Figure 3: *Identity Test*. The average accuracy of GPT-4-32k and GPT-3.5 when evaluating sums of ones with varying expression lengths and applying different symmetries. The x-axis represents the expression lengths, while the y-axis indicates the accuracy for GPT-4-32k and GPT-3.5 under various symmetry conditions. The color intensity signifies the average accuracy obtained from 10 runs for each test.

accurate results. This suggests that the model's performance may be influenced by the complexity or structure of the expressions.

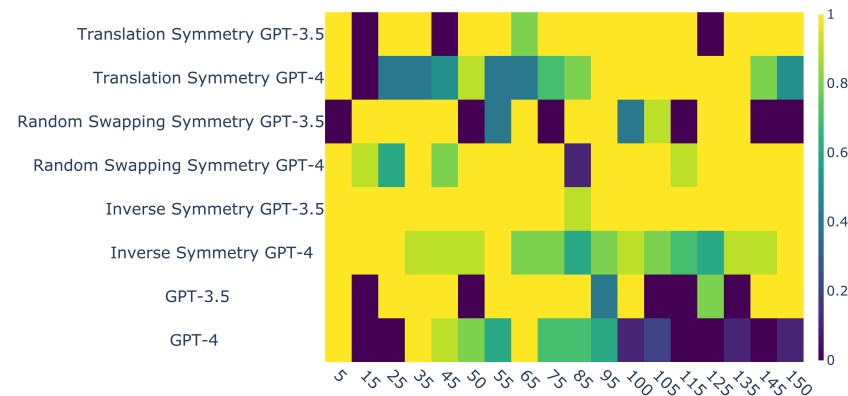

Figure 4: *Inverse Test*. The average accuracy of GPT-4-32k and GPT-3.5 when evaluating sums of ones and their inverses for various lengths. The x-axis represents the expression lengths, while the y-axis indicates the accuracy for GPT-4-32k and GPT-3.5 under various inverse symmetry conditions. The color intensity signifies the average accuracy obtained from 10 runs for each test.

## 3.6 Associativity Test

In this section we investigate the group's associativity properties. This test is crucial because if LLMs can understand associativity properties, it implies that they can simplify problems by decomposing them down into smaller components and solving them. Another benefit of analyzing this property is we can mitigate possible biases for example in a case like the summation of values near $100$ where LLMs produce an incorrect output of $100$, the associativity properties can be used to break down the problem into smaller components and mitigate the effect of biases toward these values.

For the associativity test, the expressions from the closure test have been broken down into smaller components. We have decomposed the original expression into smaller segments with ratios of $3/8$ and $5/8$, referred to as test 1, and into segments with ratios of $1/4$ and $3/4$, referred to as test 2. For each ratio, the query is divided into two segments and each segment has been passed to the LLM to obtain the result. Then, the outputs of the two segments are provided and the LLM is queried to obtain the final result. For example, for the query $1 + 1 + 1 + 1 + 1 + 1$ with test 2, we break it down into two segments: $1$ and $1 + 1 + 1 + 1$, and pass each one to the model. Suppose the LLMs' output for each query is $1$ and $4$; then, we will pass $1 + 4$ to the LLMs one last time to obtain the final result. We will repeat each experiment 10 times and report the average accuracy. Figure 5 shows the results of the associativity test. We can derive several insights from this experiment:

- Breaking down problems into smaller sub sequences improves the accuracy of LLMs.
- Segmenting and breaking down problems using test 1 yields better results for LLMs. This is reasonable since the other test creates shorter and longer segments, with the longer ones being more difficult for LLMs to solve, leading to a higher number of incorrect answers overall.
- LLMs fail to preserve associativity beyond a certain point, as they also fail the closure test.

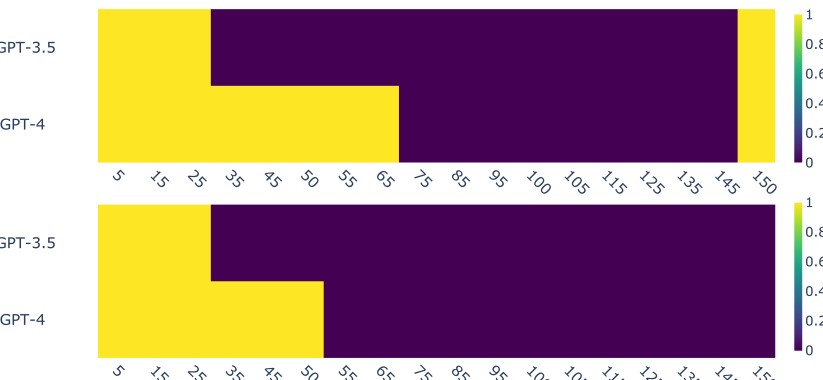

Figure 5: *Associativity Test*. The average accuracy of GPT-4-32k and GPT-3.5 for the associativity test for test 1 (top) and test 2 (bottom). The x-axis represents the expression lengths, while the y-axis indicates the accuracy for GPT-4-32k and GPT-3.5. The color intensity signifies the average accuracy obtained from 10 runs for each test.

## 3.7 Experiment on GSM8K Dataset

We also experimented with adding irrelevant information to models using the GSM8K dataset. This dataset contains grade-school math word problems, and it serves as a challenging benchmark for language models.

In this experiment, we randomly selected 200 questions from the GSM8K dataset and added irrelevant information using large language models (LLMs) such as GPT-4. To generate this irrelevant information, we provided the following prompt to the LLMs:

```
Please provide some unrelated information for the
following question.  Share 2-5 unrelated statements.
Make sure that you don't answer the question and just
provide irrelevant information.
```

For example, given the following question from the GSM8K dataset:

**Question**: Janet's ducks lay 16 eggs per day. She eats three for breakfast every morning and bakes muffins for her friends every day with four. She sells the remainder at the farmers' market daily for $2 per fresh duck egg. How much in dollars does she make every day at the farmers' market?

The irrelevant information generated by GPT-4:

**Irrelevant Information**: Janet has a passion for photography and often captures beautiful moments at the farmers' market. She is a fan of impressionist paintings and has a few replicas hanging in her home. Janet enjoys hiking in the nearby mountains during her free time.

We evaluated the performance of two models—GPT-4o and Mistral-7B-Instruct—by measuring their accuracy on the original questions and questions with irrelevant information added. The results are presented in Table 1.

| Model | Original Accuracy | Accuracy with Irrelevant Info |
|---|---|---|
| GPT-4o | 0.41 | 0.395 |
| Mistral-7B-Instruct | 0.525 | 0.48 |

Table 1: Performance of models on GSM8K dataset with and without irrelevant information.

As shown in Table 1, the performance of both models decreased when irrelevant information was added, indicating that the models struggled to filter out distractions.

## 4    CONCLUSION AND FUTURE WORK

In this study, we introduced a framework for testing the behavior of large language models based on group and symmetry principles. Our experiments involved GPT-4 and GPT-3.5, examining their performance on four group properties: closure, identity, inverse, and associativity. The results indicated that they face challenges in maintaining group properties under various circumstances. We also showed that similar behavior is observed for small language models like Llama2, Mistral, Vicuna and Phi2. The models showed significant performance drop across all tests, which might be attributed to the insufficient memory retention with respect to context. For example, after a sequence length of 15 for identity test and sequence length of 50 for closure, performance becomes $0\%$. We conducted tests with both open-weight and closed-weight models, finding that although different tokenizers may contribute to the problem, the root cause could extend beyond better tokenization strategies. A good direction for future work is exploring whether these tests can provide insights and be used as a predictor for model's performance on real-world language understanding and generation tasks.

## 5    LIMITATIONS

One key limitation is that we have not explored why models perform worse with longer text sequences or why they seem to favor certain numbers. Understanding these patterns is critical for making LLMs work better and more dependably. To address these gaps, our future work will specifically target the underlying mechanisms that contribute to performance degradation in extended sequences and the emergence of numerical biases. Moreover, it is important to expand the scope of our investigations to encompass a more diverse set of models. A comparative analysis across a spectrum of LLMs will provide valuable insights into the varied behaviors and capabilities of these models, thereby enriching our understanding of their general performance characteristics.

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

APPENDIX: EXPERIMENTS ON SLM

In the appendix we show some of our experiments with smaller language models. Our initial tests 3 presented challenges for smaller language models like Mistral-7B-Instruct. For example, in the identity test with a sequence length of five, the model had trouble generating the accurate ground truth as shown in Table 2.

| Test | Ground Truth | Mistral-7B-Instruct |
|------|--------------|---------------------|
| Identity Test | 5 | [6, 6, 7, 5, 6] |
| Translation Symmetry | 5 | [4, 5, 4, 4, 4] |
| Inverse Symmetry | 5 | [6, 6, 6, 6, 6] |
| Random Swapping | 5 | [6, 6, 6, 6, 6] |

Table 2: Mistral-7B-Instruct output for the identity test with a ground truth of 5, as described in Experiment 3.

As a result, we designed simpler tests specifically tailored for these models. In this study, we develop and release a dataset containing arithmetic questions embedded within natural language texts set in real-life scenarios. We focus on testing both smaller and larger models with this dataset, as the required reasoning involves only a few steps. The objective of the experiment is to apply group and symmetry principles to evaluate the language models' ability to solve arithmetic problems accurately. Furthermore, this experiment emphasizes the practical application of group and symmetry principles, aiming to uncover biases and limitations within these models.

The gray boxes showcase a selection of examples from this dataset.

- After creating the questions, we use the random swapping symmetry technique to shuffle the middle sentences of each question in the dataset, while keeping the first and last sentences in their original positions. By utilizing the random swapping symmetry technique, we aim to investigate its effect on the language model's performance. For each question, we generate 5 different random permutations of the sentences in the text and report the average accuracy as the performance metric. This approach allows us to assess the model's ability to adapt to varying sentence structures and maintain its problem-solving capabilities.

- In the second part of the experiment, we introduce adding irrelevant information to the questions to further evaluate the model's capabilities. For each question, we generate additional sentences, , with one example showcased in the purple box titled 'Irrelevant information' (see 5). We then shuffle the middle sentences of each question in the dataset, along with the irrelevant sentences. We provide the modified input to the model and perform five iterations to calculate the average accuracy as a performance metric. The results can be found in Table 3.

By applying the random swapping symmetry technique, we observed that the **Mistral-7B-Instruct** language model's output, indicated in blue, consistently exhibited missing or skipped parts shown in red in the same location across all shuffled versions for some of the questions in the dataset.

---

**Mistral-7B-Instruct, Version 1**

Samantha has 2 stuffed animals. Then, she received one more stuffed animal from her friend. Next, her aunt gave her one more stuffed animal. Then, she won a stuffed animal at a school event. She bought a stuffed animal from her favorite store. Later, she found one more stuffed animal in her toy box. She also bought one stuffed animal from the mall. Her brother gave her another stuffed animal. After that, her dad bought her another stuffed animal. Then she borrowed another stuffed animal from her sister. Finally, she got one more stuffed animal as a gift from her neighbor. How many stuffed animals does Samantha have now?

Mistral-7B-Instruct output: Samantha started with 2 stuffed animals. After receiving one from her friend, her aunt gave her one, she won one, bought one, found one, bought one, received one from her brother, received one from his dad, and received one from her neighbor, Samantha now has a total of 11 stuffed animals

---

**Mistral-7B-Instruct, Version 2**

Samantha has 2 stuffed animals. Then, she received one more stuffed animal from her friend. Her brother gave her another stuffed animal. Then, she won a stuffed animal at a school event. She also bought one stuffed animal from the mall. Then she borrowed another stuffed animal from her sister. Next, her aunt gave her one more stuffed animal. She bought a stuffed animal from her favorite store. Later, she found one more stuffed animal in her toy box. After that, her dad bought her another stuffed animal. Finally, she got one more stuffed animal as a gift from her neighbor. How many stuffed animals does Samantha have now?

Mistral-7B-Instruct output: Samantha started with 2 stuffed animals, then she received 1, her brother gave her 1, she won 1, bought 1, borrowed 1, received 1, bought 1, found 1, and received 1 from neighbor, making a total of 11 stuffed animals.

**Mistral-7B-Instruct, Version 1**

Emily has two puzzles. Then, she received one more puzzle from her friend. Next, her aunt gave her one more puzzle. Then, she won a puzzle at a school event. She bought a puzzle from her favorite store. Then she borrowed another puzzle from her sister. Later, she found one more puzzle in her toy box. She also bought one puzzle from the mall. Her brother gave her another puzzle. After that, her mom bought her another puzzle. Finally, she got one more puzzle as a gift from her neighbor. How many puzzles does Emily have now?

Mistral-7B-Instruct output: Emily started with two puzzles. She received one puzzle from her friend, one from her aunt, won one at a school event, bought one, borrowed one from her sister, found one, bought one at the mall, received one from her brother, and received one as a gift from her neighbor. In total, Emily now has 13 puzzles.

**Mistral-7B-Instruct, Version 2**

Emily has two puzzles. She bought a puzzle from her favorite store. Then, she received one more puzzle from her friend. Her brother gave her another puzzle. After that, her mom bought her another puzzle. Later, she found one more puzzle in her toy box. Next, her aunt gave her one more puzzle. Then, she won a puzzle at a school event. She also bought one puzzle from the mall. Then she borrowed another puzzle from her sister. Finally, she got one more puzzle as a gift from her neighbor. How many puzzles does Emily have now?

Mistral-7B-Instruct output: Emily has a total of 13 puzzles now. She got one from her favorite store, one from her friend, one from her brother, one from her mom, one from her toy box, one from her aunt, one from the school event, one from the mall, and one from her neighbor.

We observed that the Llama-2-7B-chat model exhibits similar behavior, where the language model skips some of the context of input query, leading to incorrect answers.

**Llama-2-7B-chat**

Samantha has one stuffed animal. Next, her aunt gave her one more stuffed animal. She got one more from her closest friend. After that, her dad bought her another stuffed animal. Later, she found one more stuffed animal in her toy box. Then, she won a stuffed animal at a school event. She also bought one stuffed animal from the mall. Her brother gave her another stuffed animal. Then she borrowed another stuffed animal from her sister. Then, she received one more stuffed animal from her friend. She bought a stuffed animal from her favorite store. Finally, she got one more stuffed animal as a gift from her neighbor. How many stuffed animals does Samantha have now?

Llama-2-7B-chat output: Answer: Samantha has 10 stuffed animals. Explanation: 1. She had one stuffed animal to start with. 2. Her aunt gave her one more, so she has 2 stuffed animals now. 3. Her closest friend gave her one more, so she has 3 stuffed animals now. 4. Her dad bought her another, so she has 4 stuffed animals now. 5. She found one more in her toy box, so she has 5 stuffed animals now. 6. She won one at a school event, so she has 6 stuffed animals now. 7. She bought one from the mall, so she has 7 stuffed animals now. 8. Her brother gave her another, so she has 8 stuffed animals now. 9. She borrowed

> one from her sister, so she has 9 stuffed animals now. 10. She received one more from her friend, so she has 10 stuffed animals now. Therefore, Samantha has 10 stuffed animals in total.

One implication of group and symmetry principles in this scenario is that by utilizing the output of the language model, we can identify the missing parts and potentially address inconsistencies in the output of language model which could lead to further improvements in the model's performance and a deeper understanding of the underlying patterns.

Table 3 shows the accuracy of different models on the for this dataset. Additionally, we introduce irrelevant information to each question using GPT4 model and calculate the accuracy of different models. For instance, in the case of question 5 within the box, we incorporate the following irrelevant information:

---

**Irrelevant information**

Samantha is a young girl with a passion for collecting stuffed animals. She grew up in a loving family with her parents, brother, and sister, who all supported her hobby. Samantha is known for her kind and caring nature, often sharing her stuffed animals with her friends and family. She enjoys visiting different stores and events to find unique and special stuffed animals to add to her collection. Samantha's love for stuffed animals has also inspired her to volunteer at a local children's hospital, where she brings joy to the patients by sharing her collection with them.

---

| Model | Permutation | Irrelevancy |
|---|---|---|
| Mistral-7B-Instruct | 26.6% | 18% |
| Mistral-7B-Instruct + COT | 33.3% | 24% |
| Llama-2-7B-chat | 6.89% | 1.96% |
| Llama-2-7B-chat + COT | 47.45% | 5% |
| GPT4 | 16.6% | 42% |
| GPT4 + COT | 100% | 80% |

Table 3: Comparison of language model accuracy, highlighting the impact of question permutation and the introduction of irrelevant information on model performance.

Another observation we made was that employing a chain of thought and step-by-step thinking improves the performance of the model, as demonstrated in Kojima et al. (2022). However, introducing irrelevant information to the question still negatively impacts the performance of language models like GPT-4-COT, resulting in a 20% decrease.

The accuracy of GPT-4 increased when irrelevant information was added to the question. Upon further investigation, we discovered that GPT-4, when presented with irrelevant information, generated answers that were closer to step-by-step thinking.

## APPENDIX: APPLICATION OF SYMMETRY PRINCIPLES IN GPT-4V MODEL USE CASES

In another experiment, we utilized GPT-4v to find the behaviour of GPT4-v under permutation symmetry. In this experiment, given a query figure we ask GPT4-v to compare it with another two figures using the following prompt.

---

**Prompt**

Please rate the similarity between the query figure and Figure 1 and Figure 2 using the following scale: 1 - Very similar or almost identical, 2 - Similar with minor differences, 3 - Moderately similar with noticeable differences, 4 - Not very similar with significant differences, 5 - Not similar or completely different. Provide your response as a list, with the first value indicating the similarity to Figure 1 and the second value indicating the similarity to Figure 2. Also, explain your reasoning. This is the query figure: [query.png]. Here are the two figures: [Figure1.png][Figure2.png]

---

We conducted two tests to explore GPT-4v's limitations in using group and symmetry principles. In both tests, we compared the query figure to Figure 1 and Figure 2, with one of the figures being identical to the query figure and the other being different. In the first test, Figure 1 was identical to the query figure, while in the second test, Figure 2 was identical.

We found that in first setting that Figure 1 is identical to query figure the GPT4-v response was:

> **First Setting: GPT4-v response**
>
> Figure 1 is very similar or almost identical to the query figure, with closely matching patterns, amplitudes, and frequencies of the waves and only minor variations. In contrast, Figure 2 has significant differences, displaying a distinct wave pattern with more pronounced peaks and troughs, as well as differing wave amplitudes, indicating a substantial difference in the data represented.

However, for the second test, where Figure 2 was identical to the query figure, the response was:

> **Second Setting: GPT4-v response**
>
> Figure 1 exhibits a similar overall pattern and amplitude to the query figure, with only minor differences in specific peaks and troughs. Conversely, Figure 2 has a noticeably different pattern, especially in the latter half of the figure, where the peaks and troughs do not align as closely with the query figure.

In another case, all three figures were exactly the same, and this was the response provided by GPT-4v:

> **Second Setting: GPT4-v response**
>
> Figure 1 is more similar to the query figure. Both figures have the same pattern and shape, while Figure 2 has a slightly different pattern and shape.

Upon examining figure permutations, we found that the GPT-4v model's responses can vary significantly, highlighting its sensitivity to figure arrangement. By employing symmetry and group principles, we can better understand these inconsistencies and improve the model's performance.

## APPENDIX: ABLATION STUDY

To investigate whether there is a bias towards specific points like 100 and 50, and to ensure that this is not solely related to the number 1s used in the study, we also experimented with substituting 1s with words such as 'apples', 'oranges', and 'bananas'. We asked the language models to count the number of these items within the ranges $[40 - 60]$ and $[90 - 110]$, and The predictions obtained when substituting ones with 'apples' are illustrated in Figure 6. Similar behavior was observed for other cases as well. Red dots represent incorrect predictions, while blue dots indicate correct predictions. As can be seen, the language model frequently predicts incorrectly for values around 100 and 50, exhibiting a bias towards these values.

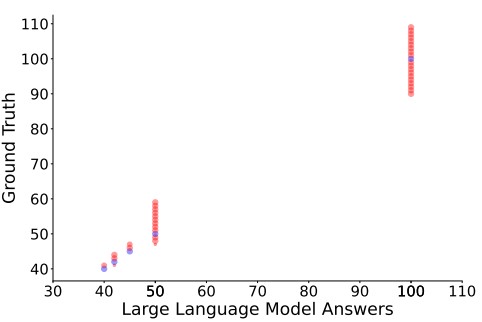

Figure 6: Scatter plot of GPT-4-32k predictions for closure test expressions using various items, illustrating the bias towards values 50 and 100. Red dots represent incorrect predictions, while blue dots indicate correct predictions. The plot demonstrates the model's tendency to predict 50 and 100 more frequently, even when the actual values are slightly different.