# OpenReview forum: "Exploring Group and Symmetry Principles in Large Language Models"
_ICLR.cc/2025/Conference — Submitted to ICLR 2025_

### Official Review · Reviewer_6eXX · 2024-10-25

**Soundness:** 3
**Presentation:** 1
**Contribution:** 2
**Rating:** 3
**Confidence:** 4

**Summary:**

The authors present different principles of symmetry and explain how those principles can be used to analyze the shortcomings of Large Language Models (LLM). The authors then provide experiments across the different principles to show how and where these LLMs fail, specifically looking at how these models perform under larger contexts over varying symmetries. All experiments are conducted on varying chains of addition between -1, 0, 1.

**Strengths:**

The authors use the known approach of analyzing symmetry within a system to probe LLMs and discover their shortcomings.

**Weaknesses:**

Main concerns:
1. The presentation of the results is extremely poor. Do not use heat maps to present accuracy. Typically you want to use line graphs for experiments that have are varying one parameter many times. Try using line graphs or bar graphs for all of the figures that are heat maps.
2. In the experimental sections, each of the results are shortly discussed in regard to how they show that the LLMs are failing. It would be useful to also discuss how these insights can lead us to build better LLMs.
3. The experimental setting of using only -1, 0, 1 makes these experiments feel like toy experiments. Using sentences and paragraphs, natural language, would be much more interesting and significant.
4. Excessive use of bullet points. Write out full paragraphs.
5. From this paper, it is still unclear how closure and associativity are different.
6. The introduction and background span 4 full pages. For a paper such as this, you could probably cut down 1 or 2 pages of content from those 4 pages so that there is more room to provide experiments and thoroughly discuss results.

Writing Comments:
Page 1, Line 41: Group and symmetry principles which made --> Group and symmetry principles have made
Page 2, Line 60 - 102: These do not need to be bullet points. The paper would be more aesthetically pleasing if the bullet points were removed and the topic markers were instead bolded. Also, Page 2, Line 58 should probably be changed to mention each of the topics instead of ending on colon.
Page 3,  Line 113 - 126: The break from bullet points and figure insertion looks odd. Keep the text with the previous bullet point and move the figure to the top of the page.
Page 4 - 5, Line 198 -222: Again, the topic markers for each bullet point can just be bolded and the bullet points can be removed.
Page 3 & 5: Should the figures presented on each page not have a caption and figure number?
Page 5, Line 269: one, zero, and negative one --> 1, 0, -1
Page 8, Line 407 - 408: Having 1.5 lines between two almost full pages figures is poor presentation.

**Questions:**

The methods are mostly clear. It is mostly a presentation and experimental scope issue. See weaknesses.

---

> ### Comment · Reviewer_6eXX · 2024-11-26
> **No response**
>
> Since there was no response from the authors, I stand by my initial review (this response was prompted by the review system).

---

### Official Review · Reviewer_ssWU · 2024-10-27

**Soundness:** 2
**Presentation:** 3
**Contribution:** 1
**Rating:** 3
**Confidence:** 4

**Summary:**

This paper takes a first-principles approach to the evaluation of language models. The authors create a succinct dataset of prompts comprising addition operations under various perturbations. The purpose of the perturbations is to evaluate whether, and the extent to which, a language model is invariant to such perturbations. The perturbations test closure, identity, inverse, and associativity properties.
 With this dataset, they report results on GPT 3.5 and 4.0 in the main body of this paper, and results on other models in the appendix. The authors also introduce irrelevant text into the prompts of GSM8K and report results on GPT-4o and Mistral-7B Instruct.

**Strengths:**

**Originality**: The first-principles approach of this paper is refreshing. Grounding a language model robustness evaluation dataset in group theoretical invariances is both clever and has the potential to provide insights beyond what is currently known about language models' behaviors.

**Quality**: The quality of the writing is overall good. It situates the work with ample and appropriate citations of related literature.

**Clarity**: The authors clearly state the purpose and contribution of the paper. The method is described clearly and illustrated nicely in the figure on p. 5.

**Significance**: Analyses of this sort have the potential to contribute valuable insights into the workings of language models and how to improve them.

**Weaknesses:**

**Originality**:  In general, papers that focus on specific operations -- like this paper, or like [1] -- demonstrate a weakness, provide valuable insights based on that weakness, and possibly explore experimentally ways to overcome the weakness. This paper demonstrates weaknesses and derives little new insights. While the concept of the dataset described in this paper is novel in ways, I'm afraid that the originality of the *insights* provided by the dataset are quite limited. The limitations of language models under conditions that require variable amounts of computation are pretty well known by now. For instance, the insight that language model accuracy on arithmetic tasks is constrained by the length of the (implied) computational graph was precisely defined and thoroughly evaluated in [1]. Their robustness under variation in the position of inputs in the context window have been explored in [2]. And the effectiveness of improving language model accuracy on such tasks by encouraging them to traverse high-probability regions of their hidden states is also well known [3]. I would encourage the authors to consider how group theoretical principles could be explored more thoroughly or systematically to provide insights beyond what has been shown already. For example, some recent work [4] has shown that recurrent networks consistently out-perform transformers as detectors of regular languages. Do similar inequalities exist between network architectures with respect to group theoretical principles?

**Quality**: The scale of the experiments reported here are small (10 runs) number of models evaluated is modest. The results of the dataset created by the authors are reported on GPT 3.5 and 4.0. The ancillary experiment described in the paper was performed on GPT-4o and Mistral-7B-Instruct. While Open AI only released their o1 model on September 12th of this year, the dataset in this paper is so small that running it against o1 could provide insight into a variable-computation model that has not yet been evaluated much (although see [5]). While simplicity is a virtue, the dataset evaluated here could be created in, I estimate, about 4 hours, so the authors could have spent much more time on other things, like evaluating group theoretical principles on more than just addition.

**Clarity**: I think the clarity of the paper is not a weakness. I would encourage the authors in a future version of the paper to put less emphasis on the potential insights that group theoretical principles can bring to language model evaluation and much more on the actual insights.

**Significance**: See my comments above under **Originality**. One point made by the authors is that the sheer smallness of their dataset implies that problems identified by other, larger datasets can be done at lower cost using their dataset. If this is indeed the case, the paper would be improved by showing at a minimum how much less expensive it is. A potentially interesting future line of research is robustness benchmarks with provably minimal cost and provable equivalence (of what is tested and surfaced) with more expensive benchmarks.

* [1] Dziri et al, Faith and Fate: Limits of Transformers on Compositionality, NeurIPS 2023
* [2] Liu et al, Lost in the Middle: How Language Models Use Long Contexts, TACL 2024
* [3] Wei et al, Chain-of-Thought Prompting Elicits Reasoning in Large Language Models, NeurIPS 2022
* [4] van der Poel et al, MLRegTest: A Benchmark for the Machine Learning of Regular Languages, JMLR 2024
* [5] Valmeekam et al, LLMs Still Can't Plan; Can LRMs? A Preliminary Evaluation of OpenAI's o1 on PlanBench, arXiv:2409.13373

**Questions:**

What insights beyond those in [1,2,3,4] cited above does your benchmark provide? What is the cost difference between using your benchmark and a set of benchmarks that surface the same weaknesses?

---

### Official Review · Reviewer_sGtk · 2024-11-01

**Soundness:** 2
**Presentation:** 3
**Contribution:** 3
**Rating:** 5
**Confidence:** 4

**Summary:**

This work examines how well LLMs adhere to principle of symmetry and group properties such as associativity and transitivity in their reasoning. These capabilities are tested by way of conducting experiments over four properties: closure, identity, inverse, and translation. These experiments reveal persistent reasoning errors ranging from bias towards certain numbers to a sensitivity to perturbations. This work also correlates these error patterns over the simple arithmetic setting to similar behaviours over more complex tasks. Finally, a synthetic dataset is proposed that contains these adversarial data samples curated to identify model reasoning pitfalls.

**Strengths:**

- There is definite value to examining the types of reasoning errors by testing independent properties to identify these patterns of behaviour in other tasks.
- This work builds on previous literature which demonstrate the susceptibility of models to simple perturbations in input data.
- There is some (qualitative) analysis comparing model outcomes on actual mathematical word problems (MWPs).
- The theoretical basis is well-researched and results in a useful framework for identifying error patterns beyond simple accuracy measures.
- There are some interesting insights re: model bias for certain numbers.

**Weaknesses:**

- The implication of the results and analyses in this work is that if LLMs can pass these property-based tests, they exhibit certain skills associated with an understanding of these properties. eg.
if this framework shows that a model has an understanding of associativity, the model can decompose and solve problems. However, there is no evidence provided for this hypothesis, for example by comparing model accuracy on the associativity test and in parallel on another pre-existing task that requires decomposition.
- Similarly, there is a lack of more comprehensive quantitative results based on accuracy metrics that confirm the hypothesis that patterns observed over the simple {1,0} addition task are also observed over more complex tasks such as solving natural language MWPs. This makes it difficult to definitively prove the claim that these tests can offer insights into the type of reasoning errors made by models.
- The {1,0} based addition test is arguably too simplistic to gauge overarching error patterns in LLMs. Furthermore, the tests proposed in this work (identity, closure, etc) could be performed on problems of far greater complexity.
- Given the insight that models are biased towards numbers such as 0, 50, 100: the inverse test is likely to yield optimistic results, as the correct answer is always 0. It may be more illustrative to perform this test with negations that amount to non-zero results, such that the sum of results over all tests in the experimental setting is zero.
- Table 3: there is no baseline performance provided over questions in their native, unperturbed state.
- Page 15: The prompt to rate similarity does not definitively define the rubric, and there is potential for ambiguity in interpreting the descriptions associated with the scores 2, 3, and 4.

**Questions:**

- Sections 3.1 and "Experiments on SLM" section of the appendix demonstrate that this framework can be utilized with any set of whole numbers. Why are the main experiments limited to elements from the set {0,1}?
- Similarly, in order to more accurately model real world problems, which contain both affirmative and negative information, it may be useful to conduct these addition-based group property evaluations with elements from the group {-1, 0, 1}.
- The translation and random swapping tests, specifically in the context of identity preservation, may result in the same input pattern. Is there any significance to or observable insight from the differing results of these tests?
- What is the intuition behind the ratios selected for Test 1 and Test 2 of the associativity test (Sec 3.6)? Providing results over a range of ratios may better identify the optimal split.
- This work proposes a variant of GSM-8K with irrelevant information in Section 3.7. How does this variant differ from existing adversarial datasets such as GSM-8K (M3) [1], GSM-8K-Adv [2], and GSM-NoOp [3], which are also perturbed variants of GSM-8K?



[1] Xie, R., Huang, C., Wang, J., & Dhingra, B. (2024). Adversarial Math Word Problem Generation.

[2] Anantheswaran, U., Gupta, H., Scaria, K., Verma, S., Baral, C., & Mishra, S. (2024). Cutting Through the Noise: Boosting LLM Performance on Math Word Problems.

[3] Mirzadeh, I., Alizadeh-Vahid, K., Shahrokhi, H., Tuzel, O., Bengio, S., & Farajtabar, M. (2024). GSM-Symbolic: Understanding the Limitations of Mathematical Reasoning in Large Language Models.

---

> ### Comment · Reviewer_sGtk · 2024-11-27
>
> Since there was no response from the authors, I stand by my initial review. I would be glad to discuss this work and amend reviews if necessary in lieu of new information provided by the authors.

---

### Official Review · Reviewer_Jgga · 2024-11-03

**Soundness:** 1
**Presentation:** 1
**Contribution:** 2
**Rating:** 3
**Confidence:** 4

**Summary:**

In this paper, the authors develop a framework for language model evaluation based on group properties and demonstrate its application to addition. Their evaluation framework centers on four key properties: closure, identity, association, and inverse. The authors reveal that GPT-3.5 and GPT-4 fail to maintain consistent performance across closure, identity, and association tests. Moreover, they show that leveraging the associative property of addition—breaking down large addition problems into smaller, more manageable pieces—improves the performance of these language models.

**Strengths:**

The symmetries and structure inherently contained in a problem can often give us a lot of information about it. I agree with the authors that this is a fundamental idea which has already permeated many areas of science. Recently, we have already seen them being applied to machine learning through many facets (e.g. geometric deep learning, neurosymbolic AI, and more recently, interpretability). Thus, the topic of the paper is timely and relevant to the field.

**Weaknesses:**

- The authors strongly emphasize that the contribution of the paper is *not* their benchmarks on arithmetic tasks but rather a general benchmarking framework. In this case, the contribution of the paper is unclear to me. Based on my understanding of the literature, it seems that the field is already aware of using  group properties/structures to improve/evaluate model performance (see [1,2,3,4,5] and much more…). To this end, many benchmarks like the authors mention themselves already incorporate these elements into their evaluation process [6].
- Given this example of arithmetic, it is unclear to me how this framework would generalize to other more complicated tasks that the community is interested in: one where the symmetries exist, but it is not at all obvious how to design benchmarks based on these symmetries. Consider examples like text-to-image, image-to-text, hallucination detection, alignment, entity resolution, etc.
- As is, the paper is very poorly written. It is hard to follow what the main claims, the experimental settings, and what even the main results are.
- The related literature section is also nonexistent/weak. I feel that the related literature should answer at least the following two questions:
    - How is this framework similar/different to other evaluation frameworks that already exist in the literature?
    - Are the experiments/results that we get from this framework consistent with what we see in the related literature (on this arithmetic task). Or even better, do the experiments/results generalize or explain some phenomena that the community does not understand?

I strongly believe this work is still in progress and would greatly benefit from substantial revisions and additional rounds of review beyond the scope of the ICLR review cycle.

[1] Godfrey, Charles, et al. "On the symmetries of deep learning models and their internal representations." *Advances in Neural Information Processing Systems* 35 (2022): 11893-11905.

[2] Liu, Ziming, and Max Tegmark. "Machine learning hidden symmetries." *Physical Review Letters* 128.18 (2022): 180201.

[3] Winter, Robin, et al. "Unsupervised learning of group invariant and equivariant representations." *Advances in Neural Information Processing Systems* 35 (2022): 31942-31956.

[4] Crabbé, Jonathan, and Mihaela van der Schaar. "Evaluating the robustness of interpretability methods through explanation invariance and equivariance." *Advances in Neural Information Processing Systems* 36 (2023): 71393-71429.

[5] Wang, Yipei, and Xiaoqian Wang. "Self-interpretable model with transformation equivariant interpretation." *Advances in Neural Information Processing Systems* 34 (2021): 2359-2372.

[6] Srivastava, Aarohi, et al. "Beyond the imitation game: Quantifying and extrapolating the capabilities of language models." *arXiv preprint arXiv:2206.04615* (2022).

**Questions:**

- In the abstract, the authors mention “assessing [language models’] reasoning capabilities remain a significant challenge. I am unaware of such challenges and the authors do not elaborate on this point in the main text further? Can you give some examples from the literature where people have shown this to be the case?
- Why do the authors call the closure test the closure test? It seems that the closure property ($a, b \in G \Rightarrow a\circ b \in G$) is not being tested, but rather the abelian property of the group (lines 231-237).
- The closure test described in lines 231-237 seems different from what the authors describe in lines 281-295. And, after reading this latter section, I no longer understand what the goal of this test is. Can the authors elaborate on this?
- Lines 441-442, for the associativity test, the authors choose only ratios (3/8, 5/8) and (1/4,3/4). Why were these specific numbers chosen and are there ablations over the “pivot” point. Did the authors test instead of splitting the problem into two distinct parts, splitting it into multiple (i.e. greater than 2)?
- Lines 455, the authors claim to that LLMs fail to preserve associativity beyond a certain point. It seems that there are multiple confounding variables here. For example, if you have length 120 then a (3/8,5/8) split corresponds to lengths (45, 75). Of these two splits, the longer one is essentially the closure test. So, why is the closure test not a subset of the associativity test?
- Based on the previous question, I guess I don’t understand what the associativity test is testing. Because you are breaking the problem down for the language model, rather than the language model recognizing that there are recursive subproblems here. So even if you demonstrate that an LLM perfectly solves this associativity test, what would you be able to say about how its generalizing?
- The authors extend their results from addition problems to the GSM8K dataset. They add irrelevant information to the prompts using GPT. How did you verify that the information being added was indeed irrelevant? I do not see any information on this in the appendix.

---

> ### Comment · Reviewer_Jgga · 2024-11-26
>
> As there were no responses from the authors, I stand by my initial review.

---

### Meta-Review · Area_Chair_UXv8 · 2024-12-20

**Metareview:**

This paper aims to develop a general framework to evaluate the reasoning capabilities of LLMs using group properties. While the topic is timely and important, all four reviewers give low scores due to the following concerns:  1) the contribution is not clear as group properties are frequently used for evaluation in the literature; 2) the presentation is poor that the literature review is insufficient and the main conclusions are not clear; 3) the experiment is limited to very simple scenarios. No rebuttal is provided from the authors. AC agrees this work is still in progress and requires substantial improvement.

**Additional Comments On Reviewer Discussion:**

No rebuttal is submitted, and so reviewers keep their initial ratings.

---

### Decision · Program_Chairs · 2025-01-22

Reject